# Stabilization of Vitamin D in Pea Protein Isolate Nanoemulsions Increases Its Bioefficacy in Rats

**DOI:** 10.3390/nu11010075

**Published:** 2019-01-02

**Authors:** Ali M. Almajwal, Mahmoud M. A. Abulmeaty, Hao Feng, Nawaf W. Alruwaili, Astrid Dominguez-Uscanga, Juan E. Andrade, Suhail Razak, Mohamed F. ElSadek

**Affiliations:** 1Department of Community Health Sciences, College of Applied Medical Sciences, King Saud University, Riyadh 11433, Saudi Arabia; dr.abulmeaty@gmail.com (M.M.A.A.); smarazi@ksu.edu.sa (S.R.); mfbadr@ksu.edu.sa (M.F.E.); 2Department of Medical Physiology, Faculty of Medicine, Zagazig University, Zagazig 44519, Egypt; 3Department of Food Science and Human Nutrition, University of Illinois at Urbana-Champaign, Champaign, IL 61801, USA; haofeng@illinois.edu (H.F.); alruwai2@illinois.edu (N.W.A.); astrid86@illinois.edu (A.D.-U.)

**Keywords:** nanoemulsion, vitamin D, vitamin D deficiency, bioefficacy, fortification, rat

## Abstract

Micronutrient delivery formulations based on nanoemulsions can enhance the absorption of nutrients and bioactives, and thus, are of great potential for food fortification and supplementation strategies. The aim was to evaluate the bioefficacy of vitamin D (VitD) encapsulated in nanoemulsions developed by sonication and pH-shifting of pea protein isolate (PPI) in restoring VitD status in VitD-deficient rats. Weaned male albino rats (*n* = 35) were fed either normal diet AIN-93G (VitD 1000 IU/kg) (control group; *n* = 7) or a VitD-deficient diet (<50 IU/kg) for six weeks (VitD-deficient group; *n* = 28). VitD-deficient rats were divided into four subgroups (*n* = 7/group). Nano-VitD and Oil-VitD groups received a dose of VitD (81 µg) dispersed in either PPI-nanoemulsions or in canola oil, respectively, every other day for one week. Their control groups, Nano-control and Oil-control, received the respective delivery vehicles without VitD. Serum 25-hydroxyvitamin D [25(OH)VitD], parathyroid hormone (PTH), Ca, P, and alkaline phosphatase (ALP) activity were measured. After one week of treatment, the VitD-deficient rats consuming Nano-VitD recovered from Vitamin D deficiency (VDD) as compared against baseline and had serum 25(OH)VitD higher than the Nano-control. Enhancement in VitD status was followed with expected changes in serum PTH, Ca, P, and ALP levels, as compared against the controls. Stabilization of VitD within PPI-based nanoemulsions enhances its absorption and restores its status and biomarkers of bone resorption in VitD-deficient rats.

## 1. Introduction

Vitamin D deficiency (VDD) remains a public health concern all over the world [1,2]. Causes of VDD include an inadequate dietary intake and limited exposure to sunlight. About 50% to 90% of vitamin D is absorbed through the skin via sunlight while the rest comes from the diet. Severe VDD leads to rickets in children and osteomalacia in adults. In adults, it also predisposes to low bone mass and contributes to bone fragility fractures in the elderly [3]. In recent years, there has been a renaissance in the study of VitD actions as evidence continues to accumulate about its role in the etiology of chronic disease such as infection response, autoimmune disease, cardiovascular disease, diabetes mellitus, and cancer [4].

Prevention of VDD remains a global health priority [1]. Food fortification with VitD is considered one of the most cost-effective strategies to combat VDD. Supplementation with VitD is also effective to address VDD [1], but this requires close control of the dose due to potential toxicity, resulting in hypercalcemia and hypercalciuria [5]. Despite the safety of food fortification, any micronutrient delivery strategy (i.e., foods vs. pills) requires evaluation in terms of compatibility with other ingredients and more importantly their efficacy in vivo. In this regard, much of the attention has been given in developing sound delivery systems to carry VitD efficiently and safely at all bio-physiological levels in the body [6]. In the last few years, lipids [7,8,9,10] and polymers [11,12,13] have been extensively investigated, used and reevaluated as prospective VitD delivery systems. 

There are certain limitations associated with these delivery systems such as short half-life, susceptibility to oxidation, possibility of hydrolysis, leakage and fusion, allergenicity, and relatively high cost of scaling up the process to achieve a reproducible, high quality products [14]. These limitations make such delivery systems less attractive to the food industry [15]. In addition to the mentioned limitations, there are some health risks associated with the consumption of cationic polymers, especially when administered at high concentrations [16]. Hence, a potential delivery system must be harmless for both the intended users (animals or humans) and the environment, in which its implementation results in minimum health risks and maximum cost-effectiveness.

One possible way to achieve this goal is by using food proteins; for example, by binding VitD to β-lactoglobulin from whey, which is a by-product of cheese making [17,18,19]. Diarrassouba et al., (2015) showed an alternative ‘green method’, which uses oppositely charged food proteins, e.g., egg lysozyme and β-lactoglobulin, resulting in protein-based microspheres with high VitD encapsulation efficiency. Most of these methods, however, involve tedious extraction procedures, gradient flows, long elution times, and often a purification step before their estimation. Moreover, several of these methods utilize solid phase extraction processes involving high processing cost [20,21]. 

Plant proteins are of particular interest as emulsifiers in food systems mainly due to their ability to adsorb to oil-water interfaces and form interfacial films [22,23]. The surface activity of proteins is the result of their amphiphilic nature, because of the presence of both hydrophobic and hydrophilic regions in their peptide chains [24]. Legume proteins are gaining popularity for this purpose due to their high natural abundance, sustainability, low cost, and functional attributes [25]. Their low solubility might reduce their use as vehicles for delivery of nutrients in foods. In a previous study, our team showed that combining ultrasound and pH-shifting treatment can enhance the functional properties pea protein isolate (PPI); specifically, higher solubility and surface hydrophobicity along with a reduction in particle size below 100 nm. As the reduction in particle size can potentiate the absorption of VitD [26], the main objective of this study was to assess the ability of novel PPI-based nanoemulsions in enhancing VitD status as evaluated in a rodent model of VDD. Our results could pave the way to develop functional PPI nanoemulsions for fortification and supplementation programs of VitD and other fat-soluble nutrients in foods. 

## 2. Materials and Methods

### 2.1. Materials

The following materials were obtained for all experiments. Pea protein isolate (PPI, NUTRALYS^®^ S85F, 85% pea protein based on dry basis) in powder form was provided by Roquette (Geneva, IL, USA), and was produced using a wet extraction process from dry yellow peas. PPI was kept at 4 °C before use. All other reagents and chemicals were purchased from Sigma-Aldrich (St. Louis, MO, USA) or Fisher Scientific (Pittsburgh, PA, USA) and were of analytical or higher grade.

### 2.2. Formation of Pea Protein Nanoaggregates

The method of Jiang et al. (2017) was followed [26]. Briefly, pea protein nanoaggregates were prepared by adding PPI (3 g) into a beaker containing 100 mL double deionized water followed by stirring for 30 min at room temperature (23 °C). PPI dispersion was adjusted to pH 12 with a few drops of 2 M NaOH. Ultrasound treatment was applied for 5 min using a VC-750 ultrasonic processor at 20 kHz (Sonics & Materials, Inc., Newtown, CT, USA). Acoustic energy was delivered to the center of the dispersion using a probe (12.5 mm diameter). Excessive heating was controlled by placing the sample on an iced water bath. After ultrasound treatment, samples were held for 1 h at room temperature before adjusting to pH 7 with a few drops of 2 M HCl. The neutralized PPI nanoaggregate dispersion was centrifuged for 15 min (Sorvall Instruments RC5C, Rotor GSA code 10, Newton, CT, USA) at 8610 RPM, 15 °C. The supernatant was collected and stored to create protein nanoemulsions. 

### 2.3. Preparation of Nanoemulsion

VitD-containing nanoemulsions were prepared by homogenizing 0.4% VitD (cholecalciferol, 98% pure) at a fixed concentration (2% *w*/*v*) of PPI-nanoaggregates as the emulsifier. Both phases were stirred for 5 min followed by ultrasound (5 min) similar as above. Samples were cooled and the final concentration as measured by reversed phase HPLC [27] was 27 µg VitD/mL.

### 2.4. Animals

All experiments with rats were conducted in the Animal Laboratory located in the College of Applied Medical Sciences (CAMS) at King Saud University. Research principles and ethical guidelines of the KSU-CAMS Research Ethics Committee were strictly observed for all experiments using animals (reference no CAMS 062-37/38). We used the model of Fleet et al. [28] to promote VDD and the delivery model of Diarrassouba et al. [11] to evaluate efficacy of delivery in rats; however for only one week to avoid any potential toxicity due to a potential enhanced absorption. Thirty-five healthy, male Wistar rats were procured from Animal Research Center, Faculty of Pharmacy, King Saud University, Riyadh, Saudi Arabia. After weaning for 3 weeks, rats were divided primarily into two groups: the control group and VitD-deficient group. The rats in the control group (*n* = 7) were fed a normal diet AIN-93G (protein 18.1%, fat 7.1%, carbohydrates 59.3%, fiber 4.8%, ash 2.2%, calcium 5.1 g/kg, phosphorus 2.8g/kg, and vitamin D 1000 IU (25 µg/kg) [29] throughout the study. The VitD-deficient rats (*n* = 28) were fed a customized VitD-deficient diet AIN-93G (the same composition of AIN-93G but with VitD < 50 IU (1.25 µg/kg) with normal Ca and P for six-weeks [28]. After six-weeks, VitD-deficient rats were randomized based on weight and 25(OH)VitD status and divided into four distinct subgroups. Treatment groups received 3 mL of 27 µg/mL VitD (1080 IU/mL) stably dispersed in nanoemulsion (Nano-VitD) or 1.8 mL of canola oil mixed with 1.2 mL delivering the same dose of VitD3 obtained from a commonly-prescribed commercially available product (70 µg/mL VitD; VIDROP, MUP Co., Egypt) (Oil-VitD). Their control groups received the same volumes of delivery vehicles, Nano-control or Oil-control, respectively. All treated groups received their doses every other day (3 doses total) within 1 week by using flexible disposable Teflon feeding tubes (Braintree Scientific, Braintree, MA, USA). During the treatment period, the estimated supply of VitD from the diet is represented in Table 1. Apart from control group, all the remaining groups continue feeding on the same VitD deficient diet which add a neglectable amount of VitD if calculated per rat during the treatment period (about 1–2 IU/rat/week). Dietary VitD intake was determined based on the measured food intake, during the treatment period, in IUs of VitD ingested by kilograms of rats per day (IU/kg/day). Figure 1 shows the experimental design, the subgroups and amounts of VitD provided.

### 2.5. Blood Sampling and Biochemical Analysis

Blood samples were collected from the lateral tail vein after six weeks of dietary conditioning. Serum samples were obtained after blood clothing at room temperature for one hour and centrifugation at 2000× *g* for 10 min at 4 °C. This blood sample was used to determine 25(OH)VitD in serum and for randomization. After one week of treatment, all animals were terminated by first using isoflurane, followed by cardiac puncture and cervical dislocation. Blood collected at this point was used for the determination of 25(OH)VitD, parathyroid hormone (PTH), Ca, P and alkaline phosphatase activity (ALP). Right and left femur bone was used to prepare histopathological sections. Plasma was separated by centrifugation and stored at −80 °C until analysis.

Serum 25(OH)VitD and PTH assays. 25(OH)VitD and PTH concentrations in serum were determined using ELISA kits (MyBio-Source, San Diego, CA, USA; #MBS2601819 & #MBS265580, respectively). Serum 25(OH)VitD and PTH for all specimens and controls were conducted concurrently with the standards, and concentrations were calculated from external standards. The coefficient of variance of the 25(OH)VitD and PTH were 10 and 20%, respectively.

Ca, P, and ALP. Ca (#MBS8243246) and P (#MBS8243207) concentration and alkaline phosphatase activity (ALP) (#MBS2540468) in serum were analyzed using colorimetric kits following the steps of the manufacturer’s instructions provided along with the kits (MyBio-Source, San Diego, CA, USA). The inter-day coefficient of variation for each of the assays was 10, 10, and 20% for Ca, P, and ALP, respectively.

### 2.6. Histological and Histomorphometric Analysis

The femur specimens from all rats were collected. Femur samples of the control group were labeled as ‘controls’. All specimens were decalcified, sectioned, processed and stained by Hematoxylin, and Eosin. Gross and microscopic description were done for all femur samples and they were rated according to the criteria as mild, moderate or severe osteoporotic changes [30]. The histomorphometric analysis of the selected regions of interest (ROIs) was done by Image J software (Image J, National Institutes of Health, Bethesda, MD, USA). Tools of Image J were used to outline and calculate the osteoid area and the area of bone marrow separating the bone trabeculae (trabecular separation) [31]. Two ROIs from each rat sample were selected for analysis and the presented as mean ± SD.

### 2.7. Statistical Analysis

Data were expressed as mean ± SD. The normality of distribution for the study variables was assured by Kolmogorov–Smirnov test. All variables followed a normal distribution since all *p* values > 0.05. Before and after effects of dietary treatments on 25(OH)VitD was evaluated using paired *t*-test. ANOVA with Tukey HSD post hoc test were used to compare group effects on all variables. All differences were considered significant at an alpha of 0.05. All statistical analyses were conducted using the SPSS for Windows (version 19.0; SPSS Inc., Chicago, IL, USA).

## 3. Results

### 3.1. Changes of 25(OH)VitD and Biomarkers of VitD Deficiency Effects on Bone

Table 1 shows levels of 25(OH)VitD in serum for all study groups before and after one-week of treatment. After one week of treatment, VitD deficient animals receiving Nano-VitD showed a higher circulating levels of VitD (34.38 ± 7.00 nmol/L) than before the treatment (14.65 ± 1.29 nmol/L) or the Nano-control (15.88 ± 5.77). Rats receiving VitD mixed in canola oil showed no improvement in VitD status (*p* > 0.05). There were no differences in animal body weight after seven weeks of experiments (Table 1). At the end of the treatment, provision of Nano-VitD restored the levels of biomarkers of VDD as compared to the Nano-control, and these levels were similar to those in the control group (Table 2).

### 3.2. Histological Changes after Treatment with VitD Deficient Diets and Therapies

Gross examination of cut sections after six-weeks of VitD-deficient diet revealed no significant histopathological changes. However, as shown in Figure 2, microscopically regions of interest (ROIs) of examined sections revealed rat bone tissue showing reduced thickening of the bone cortex (osteoid area) associated with widely separated bone trabeculae (trabecular separation) containing bone marrow element.

After one week of dietary treatment, a significant improvement of the osteoid area rather than trabecular separation of the bone sections was observed by histomorphometric parameters (Table 3) of Image J software in the Nano-VitD group and its deficient counterpart as shown Figure 3. As can be seen in Figure 4, however, consumption of VitD in canola oil failed to produce a significant improvement in the osteoid area compared with its counterpart. An increase in trabecular separation was observed in animals receiving canola oil with VitD.

## 4. Discussion

In the present study, we show that consumption of VitD stably dispersed in a PPI nanoemulsion, created by sonication and pH shifting combined treatment as shown in our previous work [26] resulted in improved VitD status and a near complete recovery from the symptoms of vitamin D deficiency (VDD). After one week of treatment, only the group receiving VitD (as cholecalciferol) dispersed in PPI nanoemulsions showed improvement in 25(OH)VitD, the circulating form of VitD and biomarker of its status [32]. Moreover, animals receiving the Nano-VitD treatment showed an expected directional change in all blood biomarkers of bone turnover indicative of recovery from deficiency (e.g., lower PTH and ALP, and higher Ca and P) closer to the levels found in the VitD control receiving the normal diet. The increase in serum 25(OH)VitD was 2.3 times higher than its control (Nano-control) after one-week of treatment, potentially due to increased protection of the vitamin during digestion [27] and enhancement of oral bioavailability due to improved micellarization of VitD in the small intestine [33,34]. The group receiving Oil-VitD did not improve 25(OH)VitD status. This could be due to lower absorption. The VIDROP product disperses cholecalciferol in β-cyclodextrin and polysorbate 20. According to the vendor’s recommendations for treatment of florid rickets, the product should be consumed daily for at least 3 weeks. Thus, it is possible that the short treatment period reduced its efficacy. In a recent study, Kadappan et al., (2018) created vitamin D-containing nanoemulsions using Q-NATURALE^®^ 200 V, an extract from *Quillaia Saponaria* Molina, as the surfactant and showed enhanced micellarization of VitD and absorption in rats [34]. Moreover, after single dose administration of nanoemulsified lipophilic vitamin mixture (i.e., vitamins A, D, E) to rats showed higher plasma area under the curve concentration for these vitamins compared to controls [35]. Though these studies inform on the bioavailability of vitamin D in nanoemulsions, it is important to consider its functionality and efficacy.

Nutrient depletion/repletion models in animals are useful to study the functional impact of new delivery vehicles that can be added into the food supply. We used the model of Fleet et al., (2008) to promote VDD in rats [28] and provided the treatment based on studies conducted previously [11]. The model used growing rats to accelerate vitamin D deficiency. We chose a short period of treatment as the dose used (~0.32 mg/kg body weight (bw)) results in a significant increase in VitD status [36]. Oral supplementation of VitD dispersed in nanoparticles in water increases circulating levels within three days [34]. Thus, to avoid any potential toxicity, in this study we gavaged the animals every other day within one week (3 doses of 81 µg VitD each). This dose is higher than the lowest observed adverse effect level (LOAEL = 0.06 mg/kg bw) but significantly lower than the LD50 (35 mg/kg bw) for male rats [37]. Though our interest is on using this technology in foods, we focused this first step on efficacy using a dose that reflects supplementation rather than fortification. This strategy helped remove potential VitD interactions with the food matrix, which might limit the interpretations of our results. 

Though there were no changes in body weight, serum 25(OH)VitD levels were markedly different between the control and deficient groups before randomization. Vitamin deficiency was ascertained by attaining low levels of 25(OH)VitD as well as changes in bone turnover biomarkers such as increased PTH signaling (secondary hyperparathyroidism) and alkaline phosphatase activity in blood. The expected trend changes due to VDD were similar, though not to the same extent, to those found in previous studies with VDD in rats [28,38]. PTH is an important hormone for bone metabolism, maintaining the normal serum concentrations of calcium and phosphorous. Increased PTH concentrations will lead to an increase in bone turnover, causing negative bone balance and increased fracture risk [39]. Deficiency of VitD results in secondary hyperparathyroidism, a symptom common in rickets and osteomalacia [40]. Moreover, ALP plays a role in bone mineralization and phosphorus homeostasis [41]. It is has been shown to increase during osteomalacia and rickets [42,43]; however it is not a reliable biomarker of vitamin D deficiency [44]. 

VitD is important in regulating homeostasis of Ca and P, and severe VDD is known to increase the risk of bone and kidneys [45]. The active or hormonal form of VitD [1,25(OH)_2_VitD] binds to the VitD receptor in the nucleus and triggers a series of effects in several tissues including the kidney, the bone, and the intestines, all which increase serum Ca [46,47]. Although VitD interacts with PTH signaling [46], hypovitaminosis D does not influence PTH under sufficient serum levels of Ca. In the kidneys, VitD is regulated by PTH, Ca and P levels [48]. Moreover, hypophosphatemia was observed in all groups except the control as well as the Nano-VitD group. It is well known that a certain degree of phosphate depletion may develop in VDD [49]. Renal transport of Ca is known to be affected by phosphate transport. It has been reported that the depletion of phosphate is related to hypercalciuria [50,51] with a resistance to PTH actions to stimulate tubular reabsorption of Ca [47], to inhibit tubular reabsorption of phosphate [52], and consequently raises serum Ca in blood. The occurrence of hypercalciuria by phosphate depletion has been attributed to the increase in the filtered load of Ca and combined with the elevation of ALP concentrations in VDD groups. The Nano-VitD group showed normal VitD levels, which may be due to accompanying improvement of calcium level, and the decrease in the tubular reabsorption due to secondary amelioration of the functional hypoparathyroidism [53,54]. Also, Coburn and Massry, (1970) have suggested that hypophosphatemia can alter renal handling of Ca and reduces the responsiveness to PTH [50]. High levels of PTH were remarked with adverse effects on bone resorption and low bone mass [55]. The mature osteoclast removes calcium and phosphorus from the bone to maintain blood calcium and phosphorus levels. Adequate calcium and phosphorus levels promote the mineralization of the skeleton [32].

Nanoemulsions are an alternative option to deliver lipid-soluble nutrients and bioactive compounds [56]. Nanoemulsions are colloidal dispersions that contain small particles (20–200 nm) amenable for the dispersion of lipophilic substances such as fat-soluble vitamins dispersed within an aqueous medium and known to increase their oral bioavailability [33,57,58]. This study is consistent with those findings of others showing that reduction of particle size of liposoluble bioactives and their stabilization using different surfactants results in their increased absorption [33,57,58]. For instance, nanoemulsification has increased the oral bioavailability in rats of anti-inflammatory drugs emulsified with Tween 20 [59] or anti-cancer bioactives emulsified with Cremorphor EL and polyethylene glycol [60], bioactives such as Kenaf seed oil emulsified with sodium caseinate and Tween 20 [61], vitamin E emulsified in lecithin [62], and coenzyme Q10 emulsified with salmon lecithin [63]. In the case of food applications and due to public pressure for clean labels, the food industry avidly seeks food-grade, natural surfactants, which are usually made of proteins, polysaccharides, or their combination [64,65]. In this study, stabilized PPI was used as a suitable protein-based surfactant with the ability to disperse and protect VitD [27,66]. PPI is an intriguing surfactant as it is plant-based, hypoallergenic, possesses high hydrophobicity, and is commercially available [9,67].

Although to a great extent eradicated from many countries, VDD remains a public health concern all over the world [1,2,3]. Prevalence rates of VDD and insufficiency are high among all age groups, where women and children are most vulnerable groups [2,3,68,69,70]. Infants are at a higher risk of VDD or VitD insufficiency if they are born from young mothers who have had multiple pregnancies, and are non-White race/ethnicity [71]. Non-White populations (skin type V) are at a higher risk of VDD and VitD insufficiency because they are unable to efficiently synthesize VitD in the skin [72,73]. In Middle Eastern countries, where excessive heat, life style choices, and cultural norms reduces sunlight exposure of populations groups, VDD and insufficiency are highly prevalent [74]. Moreover, VitD is difficult to obtain from the ordinary diet because it is not naturally present in many foods. Thus, food fortification with VitD has been proposed as a nutrition specific strategy with the widest reach and impact in the population in terms of enhancing VitD status [6]. 

Current fortification technologies are limited to a few target foods (e.g., oil and dairy products). This is due to the high sensitivity of VitD to oxidation, UV light, and temperature as well as the potential deleterious effects of new technologies on food attributes (e.g., color, flavor), which could affect consumer compliance [75]. Despite the existence of, albeit limited, fortified products, fortification strategies have had limited impact in several countries due to regulatory, monitoring and evaluation issues [76], limited number of suitable foods for VitD fortification [77], and most importantly, lack of culturally appropriate fortification delivery systems. Our study addresses these identified problems by proposing the use of nanoemulsions to disperse and protect VitD within foods as well as enhancing its absorption. The results of the present study will help in devising a strategy that may increase VitD efficacy as a fortificant and hence assist in global efforts to reduce VDD. 

Finally, our results should be considered with caution. The model used does not allow a full understanding of the bioavailability or pharmacokinetics of VitD from either source. This model and the form of delivery of VitD allows for understanding of its bioefficacy. Nonetheless, it is not a model of fortification, in which lower doses for longer times are used. The large dose and study length used was based on a previous study that showed increased absorption of vitamin D in rodents within 3 weeks of oral delivery [11,34]. It is possible that this short treatment period influenced the efficacy of the two sources of VitD evaluated and their effects on serum 25(OH)VitD and bone parameters. Though this study used a large dose of VitD (as cholecalciferol, 0.32 mg/kg bw) this dose was below the LD50 for male rats (35 mg/kg bw) and within a short period of time [78]. PPI can be used to encapsulate other fat-soluble micronutrients and bioactives needed in fortification and supplementation programs such as vitamin A, E, and carotenoids. Nonetheless, the interaction of PPI alone and carrying nutrients with other ingredients and micronutrients within a more complex food matrix requires further evaluation. In addition, the potential influence of PPI carrying micronutrients on the sensory characteristics of foods is an important aspect that will dictate the success of this technology in food fortification programs. Apart from VitD, the measurement of baseline biochemical parameters was missed in this study. This was a limitation for this investigation. 

## 5. Conclusions

The evidence from this study suggests that the consumption of VitD dispersed in PPI nanoemulsion created by ultrasound and pH shifting improved its circulating status, which instead influenced the levels of bone turnover biomarkers in VitD-deficient rats. Future studies should focus on evaluating the potential toxicity of VitD delivered in PPI nanoemulsions as well as determining their potential effects on foods. 

## Figures and Tables

**Figure 1 nutrients-11-00075-f001:**
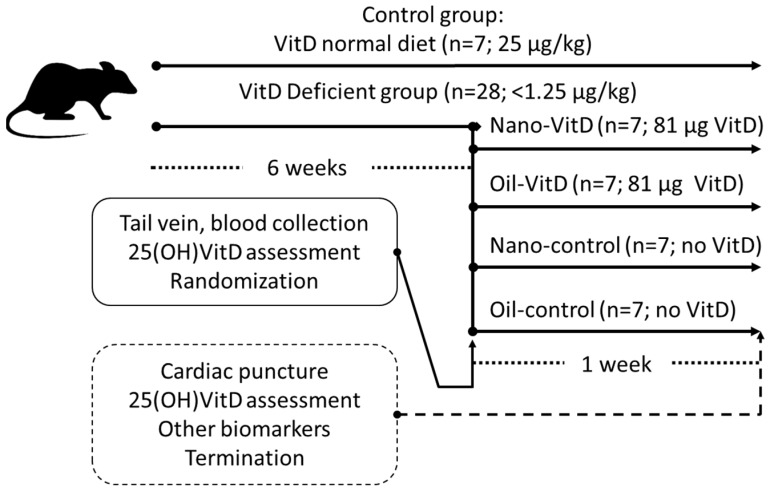
Study experimental design.

**Figure 2 nutrients-11-00075-f002:**
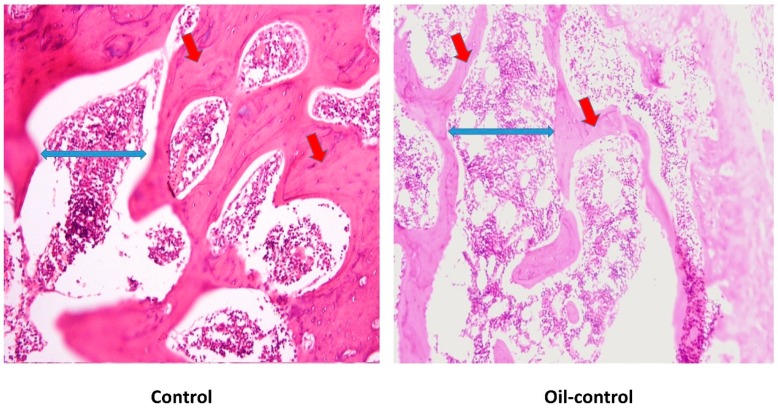
Comparison of histological sections of rat’s femur from control group (**A**) vs. Oil-control group (**B**) at the end of the study. Red arrows are plotting the osteoid area and the blue ones referring to the trabecular separation.

**Figure 3 nutrients-11-00075-f003:**
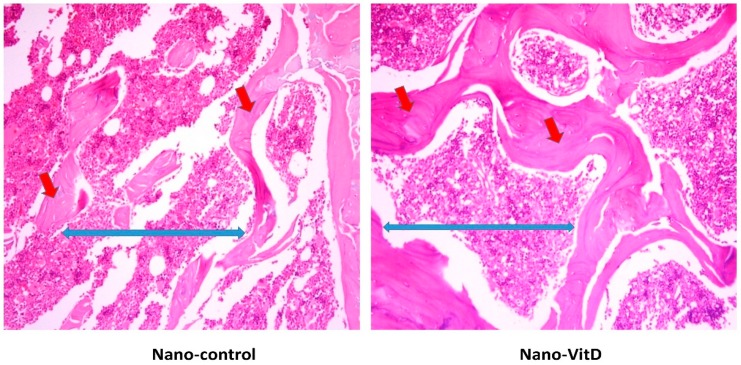
Histopathological changes in femur of rats receiving Nano-VitD vs. Nano-control after one week of treatment. Red arrows are plotting the osteoid area and the blue ones referring to the trabecular separation.

**Figure 4 nutrients-11-00075-f004:**
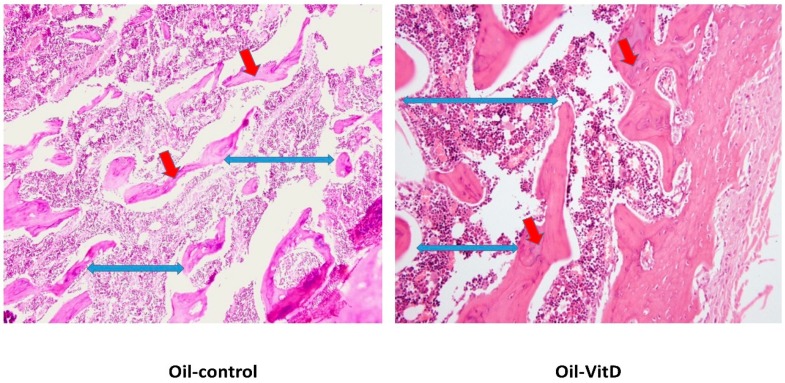
Histopathological changes in femur of rats receiving Oil-VitD vs. Oil-control after one week of treatment. Red arrows are plotting the osteoid area and the blue ones referring to the trabecular separation.

**Table 1 nutrients-11-00075-t001:** Rat body weights and levels of 25(OH)VitD in serum before and after one week of treatment.

		Body Weight Changes (g)		25 (OH)VitD (nmol/L)	
Groups ^1^	Dietary VitD Intake (IU/kg/day) ^1^	Before	After	Before	After
Control	57.78 ± 6.49 ^a^	247.76 ± 29.47	262.28 ± 36.98 *	31.68 ± 10.40 ^a^	36.84 ± 9.16 ^a^
Nano-control	1.32 ± 0.11 ^b^	258.12 ± 21.90	270.45 ± 31.72	15.38 ± 5.51 ^b^	15.88 ± 5.77 ^b^
Oil-control	1.30 ± 0.35 ^b^	225.86 ± 21.56	249.48 ± 26.96	18.26 ± 6.38 ^b^	15.59 ± 2.45 ^b^
Nano-VitD	1.46 ± 0.18 ^b^	239.06 ± 26.10	253.11 ± 26.64	14.65 ± 1.29 ^b^	34.37 ± 7.00 ^b,^*
Oil-VitD	1.46 ± 0.12 ^b^	256.24 ± 25.09	266.13 ± 27.53	14.33 ± 3.43 ^b^	14.05 ± 3.08 ^a^

^1^ Animal groups receiving: Control (VitD normal diet), VitD dispersed in nanoemulsion (Nano-VitD), VitD mixed in oil (OilVitD), nanoemulsion without VitD (Nano-control), and canola oil without VitD (Oil-control). Results are presented as Means ± SD. The asterisk (*) indicates significant differences (*p* < 0.05) after paired t-test evaluating before and after effects (within rows). When present, different superscripts within each column represent statistical differences after One-way ANOVA and Tukey’s Honest Significant Difference (HSD) test (*p* < 0.05).

**Table 2 nutrients-11-00075-t002:** Concentration of several blood biomarkers of VitD deficiency in rats after one week of dietary treatments.

Groups ^1^	PTH ^2^ (pg/mL)	Ca (mg/dL)	P (mg/dL)	ALP (U/L)
Control	23.36 ± 12.00 ^a^	10.24 ± 0.92 ^a^	3.67 ± 1.13 ^a^	58.5 ± 11.5 ^a^
Nano-control	37.54 ± 6.61 ^a^	7.12 ± 1.16 ^b^	1.38 ± 0.57 ^b^	196.2 ± 57.7 ^b^
Oil-control	78.93 ± 8.31 ^b^	6.68 ± 1.92 ^b^	1.17 ± 0.62 ^b^	171.0 ± 17.6 ^b^
Nano-VitD	25.22 ± 14.26 ^a^	9.64 ± 0.60 ^a^	3.65 ± 0.71 ^a^	72.4 ± 31.0 ^a^
Oil-VitD	86.05 ± 9.67 ^b^	5.32 ± 1.28 ^b^	1.33 ± 0.32 ^b^	182.6 ± 61.8 ^b^

^1^ Animal groups receiving: Control (VitD normal diet), VitD dispersed in nanoemulsion (Nano-VitD), VitD mixed in oil (Oil-VitD), nanoemulsion without VitD (Nano-control), and canola oil without VitD (Oil-control). ^2^ PTH: parathyroid hormone; ALP: alkaline phosphatase. Results are presented as Means ± SD. When present, different superscripts (^a,b^) within each column represent statistical differences after One-way ANOVA and Tukey’s HSD test (*p* < 0.05).

**Table 3 nutrients-11-00075-t003:** Histomorphometric parameters among study groups. When present, different superscripts within each column represent statistical differences after One-way ANOVA and Tukey’s HSD test (*p* < 0.05).

Groups ^1^	Osteoid Area (mm^2^)	Trabecular Separation (mm^2^)
Control	7.67 ± 0.91 ^a^	1.18 ± 0.51^a^
Nano-control	4.42 ± 1.49 ^b^	2.31 ± 1.11 ^b^
Oil-control	4.13 ± 1.06 ^b^	0.74 ± 0.48 ^a^
Nano-VitD	6.36 ± 1.16 ^a^	2.55 ± 1.39 ^b^
Oil-VitD	4.41 ± 1.09 ^b^	2.13 ± 1.18 ^b^

^1^ Animal groups receiving: Control (VitD normal diet), VitD dispersed in nanoemulsion (Nano-VitD), VitD mixed in oil (Oil-VitD), nanoemulsion without VitD (Nano-control), and canola oil without VitD (Oil-control).When present, different superscripts (^a,b^) within each column represent statistical differences after One-way ANOVA and Tukey’s HSD test (*p* < 0.05).

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
