# Peer review of "Stabilization of Vitamin D in Pea Protein Isolate Nanoemulsions Increases Its Bioefficacy in Rats"

_nutrients, 2019, doi:10.3390/nu11010075_

Round 1
Reviewer 1 Report
The manuscript entitled “Stabilization of Vitamin D in Pea Protein Isolate Nanoemulsions Increases its bioefficacy in Rats” presents interesting issue, but some important corrections are necessary.
General:
The manuscript should be prepared according to the instructions for Authors and some aspects must be corrected – e.g. Abstract (“The abstract should be a single paragraph and should follow the style of structured abstracts, but without headings”), references (doubled numbering should be removed), etc.
“Control sufficient group” – the other description of group is needed, as a word “sufficient” does not present properly the characteristics – they were rather with no vitamin D deficiency, not “sufficient”.
The treatment should be briefly defined for groups: Nano+VitD, Oil+VitD, Nano-VitD, Oil-VitD – especially if for Nano-VitD, Oil-VitD, the “minus” sigh looks like “hyphen” sigh, so Authors should think about changing the names to obtain better readability
Authors should correct the wording – e.g. they should not indicate that “vitamin D deficient rats […] recovered from VitD deficiency […] compared to the sufficient control”, as they may have either “recovered from VitD deficiency […] compared to the baseline results” or “be characterised by higher VitD level […] compared to the sufficient control” – it should be corrected accordingly in the whole manuscript
Abstract:
Background – Authors should properly justify the study – not only present the aim, but aim accompanied by proper justification
The vitamin D content in the diets should be presented and the intake should be compared between groups.
Introduction:
Lines 35-43 – Authors present a number of basic or even trivial information that are well known for the readers of the Nutrients journal – such information must be reduced
The section must be more consistent with the conducted study – Authors assessed the model of young rats – rather the model of rickets than osteomalacia, so more information about rickets is needed and not about osteomalacia
Authors must briefly introduce the aim of the study – it is not known what model is analysed – the model of fortified food products (if so, Authors did not present the influence of nutrients present in the fortified products that may interact) or the model of supplementation (if so, why the Introduction section presents the issue of food fortification).
The rest of the section should be corrected accordingly – e.g. if the model of food fortification is analysed – what products are in general fortified, what amount of vitamin D is added, what is the share of fortified products in vitamin D supply, etc.
Materials and methods:
Lines 105-107 – The specific number of ethics committee agreement should be presented (referred), as on the basis of the presented info it is not explicit, that the research obtained the agreement to be conducted (“Research principles and ethical guidelines of the KSU-CAMS Research Ethics Committee were strictly observed for all experiments using animals”).
Was the baseline status analysed or just after intervention?
Line 112 – “This diet contained 1,000 IU (25 μg) of cholecalciferol per kg.” the information is reproduced from the previous sentence
The vitamin D intake (based on the intake of diet) must be presented and compared.
It seems, that Authors did not verify the normality of distribution for the assessed variables. Authors must verify the normality of distribution and specify the test applied for verification.
If the distribution is normal, the mean values should be presented (accompanied by SD), but if it is different than normal, the median, accompanied by minimum and maximum values should be presented – it should be specified that distribution is normal if it is.
The applied statistical tests should be corrected accordingly based on distribution.
Results:
If the distribution is normal, the mean values should be presented (accompanied by SD), but if it is different than normal, the median, accompanied by minimum and maximum values should be presented – it should be specified that distribution is normal if it is.
Table 1 – Authors should compare the baseline vitamin D levels for “VitD” groups
The baseline body mass for groups should be presented and compared
Figure 3(B), Figure 4(B) – should be presented rather as tables to be easier to follow
Discussion:
Authors should address the short period of the study – Authors analysed the results after only one week of intervention – why such a short intervention was planned and how it may have influenced the observed results?
It seems that a number of factors were not analysed at baseline, but just after intervention – it should be explained and discussed.
The limitations of the study should be extensively discussed
Authors should address the previously indicated issues associated with the applied model
Conclusions:
The potential mechanism was not studied, so it should not be addressed in the conclusion
Author Contributions:
The section is shabbily prepared
X.X. (line 303) – who is he?
H.F. – it seems that he participated only in resources (what do Authors mean by “resources”?) and funding acquisition
S.R. - it seems that he participated only in investigation (what do Authors mean by “investigation”? animal experiment?)
M.F.E. – it seems that he did not participated in the study at all
Authors who did not participate in the manuscript preparing should be removed (due to the risk of guest authorship procedure that is forbidden) and presented in Acknowledgements Section
Reviewer 2 Report
The main aim of the paper is to evaluate the bioefficacy of vitamin D (VitD) encapsulated in nanoemulsions developed by sonication and pH-shifting of pea protein isolate in restoring VitD status in VitD-deficient rats.
The study shows interesting results and the data is well analysed, but discussion and description of the method should be improved.
Specific comments
Line 18. Treatment and control groups are named in the same way?
Line 35. More references are needed in this paragraph.
Line 76. Nanoemulsions and this procedure should be explained and referenced
Line 107. Why only males?
Line 112. Composition of the diet for VitD-deficient rats? Fiber, fat…
Line 115. How the rats received the VitD? Included in the food, injected in the mouth? Later is said by gavage, but how?
Line 116. “1.8 mL of canola oil followed by 1.2 mL” explain better. What does it mean? How was determined the quantity of vitD in canola oil?
Line 117 “VitD3” What does it means?
Line 295. Economic analysis is missing or at least a discussion of the cost of implement this approach in developed or non-developed countries.
Round 2
Reviewer 1 Report
The manuscript entitled “Stabilization of Vitamin D in Pea Protein Isolate Nanoemulsions Increases its bioefficacy in Rats” presents interesting issue, but some important corrections are necessary.General:
“Control sufficient” (line 219) – the other description of group is needed, as a word “sufficient” does not present properly the characteristics.
The treatment should be briefly defined for groups: Nano+VitD, Oil+VitD, Nano-VitD, Oil-VitD – especially if for Nano-VitD, Oil-VitD, the “minus” sigh looks like “hyphen” sigh, so Authors should think about changing the names to obtain better readability. – I understand that it is defined in Figure 1, but please think about a situation when reader of the journal finds the abstract – he does not know that he must start from figure 1 to understand the abstract properly.
Authors should correct the wording – e.g. they should not indicate that “after one week of treatment, the VitD-deficient rats consuming Nano +VitD recovered from VitD deficiency as compared against baseline values and the deficient control consuming Nano –VitD” – first part of the sentence is proper, but the second is not – there may be higher values as compared against deficient control, but the animals didi not “recover […] as compared against […]deficient control” – it should be corrected accordingly in the whole manuscript
Abstract:
Background – Authors should properly justify the study – not only present the aim, but aim accompanied by proper justification – Sometimes Abstract may be a little bit longer in order to be properly understood by readers.
The vitamin D content in the diets should be presented in the Abstract and the intake should be compared between groups.
Introduction:
Authors must briefly introduce the aim of the study – in a simple sentence (e.g. “the aim of the study was…”)
It is still confusing that Authors are mixing the issue of food fortification and supplementation. It seems that Authors want to present all the information that they have about vitamin D with no criticism. Authors must briefly present what model did they apply – the model of fortified food products (if so, Authors did not present the influence of nutrients present in the fortified products that may interact) or the model of supplementation (if so, why the Introduction section presents the issue of food fortification). Afterwards the presented information must be associated with the direct aim and the model.
Materials and methods:
If the distribution is normal, Authors should state it.
Results:
Table 1 – Authors should indicate not only dietary vitamin D intake but also total supply – from diet with supplementation added.
Figure 3(B), Figure 4(B) – should be presented rather as tables to be easier to follow
Discussion:
Authors should extensively address the short period of the study – Authors analysed the results after only one week of intervention – they should discuss why such a short intervention was planned and how it may have influenced the observed results
It seems that a number of factors were not analysed at baseline, but just after intervention – it should be extensively explained and discussed.
Author Contributions:
The section is shabbily prepared
X.X. (line 304) – who is he?
M.F.E. – it seems that he did not participated in the study at all
Authors who did not participate in the manuscript preparing should be removed (due to the risk of guest authorship procedure that is forbidden) and presented in Acknowledgements Section
Author Response
Open Review
(x) I would not like to sign my review report
( ) I would like to sign my review report
English language and style
( ) Extensive editing of English language and style required
( ) Moderate English changes required
(x) English language and style are fine/minor spell check required
( ) I don't feel qualified to judge about the English language and style
Yes | Can be improved | Must be improved | Not applicable | |
Does the introduction provide sufficient background and include all relevant references? | ( ) | ( ) | (x) | ( ) |
Is the research design appropriate? | ( ) | (x) | ( ) | ( ) |
Are the methods adequately described? | ( ) | (x) | ( ) | ( ) |
Are the results clearly presented? | ( ) | (x) | ( ) | ( ) |
Are the conclusions supported by the results? | (x) | ( ) | ( ) | ( ) |
Comments and Suggestions for Authors
Dear Reviewer,
Thanks for taking the time to offer a fair review. We have bolded your comments in black for easy reading. Our responses are in red in this document and in the revised version of the manuscript. We hope this review helps clarify your questions.
The manuscript entitled “Stabilization of Vitamin D in Pea Protein Isolate Nanoemulsions Increases its bioefficacy in Rats” presents interesting issue, but some important corrections are necessary.
General:
“Control sufficient” (line 219) – the other description of group is needed, as a word “sufficient” does not present properly the characteristics.
We have amended this. We have changed defined the group of animals consuming the normal diet with recommended amounts of vitamin D as the Control group.
The treatment should be briefly defined for groups: Nano+VitD, Oil+VitD, Nano-VitD, Oil-VitD – especially if for Nano-VitD, Oil-VitD, the “minus” sigh looks like “hyphen” sigh, so Authors should think about changing the names to obtain better readability. – I understand that it is defined in Figure 1, but please think about a situation when reader of the journal finds the abstract – he does not know that he must start from figure 1 to understand the abstract properly.
We agree and sorry for the confusion. We used a different system for nomenclature. Hope this is helpful.
Authors should correct the wording – e.g. they should not indicate that “after one week of treatment, the VitD-deficient rats consuming Nano +VitD recovered from VitD deficiency as compared against baseline values and the deficient control consuming Nano –VitD” – first part of the sentence is proper, but the second is not – there may be higher values as compared against deficient control, but the animals didi not “recover […] as compared against […]deficient control” – it should be corrected accordingly in the whole manuscript
We have changed this sentence in the abstract.
Abstract:
Background – Authors should properly justify the study – not only present the aim, but aim accompanied by proper justification – Sometimes Abstract may be a little bit longer in order to be properly understood by readers.
Sorry for our misunderstanding. We have added justification for this study in the abstract. Reduction in particle size and its stabilization in nanoemulsions have been shown to increase the absorption of nutrients and bioactives. Nonetheless, their efficacy, beyond increase in absorption, has to be evaluated in vivo.
The vitamin D content in the diets should be presented in the Abstract and the intake should be compared between groups.
The abstract of the previous review includes this information. “Weaned male albino rats (n=35) were fed either normal diet AIN-93G (VitD 1000 IU/kg) (control group; n=7) or a VitD-deficient diet (<50 IU/kg) for six weeks (VitD-deficient group; n=28).” The intake is included in Table 1. The comparison is given between groups as well. Also, the animals received a gavage dose, so it was not through dietary means per se (daily eating the diet), but orally.
Introduction:
Authors must briefly introduce the aim of the study – in a simple sentence (e.g. “the aim of the study was…”)
The following is verbatim from the previous review. “the main objective of this study was to assess the ability of novel PPI-based nanoemulsions in enhancing VitD status as evaluated in a rodent model of VDD”. Please indicate if this is not simple enough.
It is still confusing that Authors are mixing the issue of food fortification and supplementation. It seems that Authors want to present all the information that they have about vitamin D with no criticism.
We are not mixing this issue, either strategy (food fortification vs. supplementation) requires evaluation of efficacy. We are using a model that allow us to test this in the short term. Though We will follow this study with a longer term study that models better food fortification. .
Authors must briefly present what model did they apply – the model of fortified food products (if so, Authors did not present the influence of nutrients present in the fortified products that may interact) or the model of supplementation (if so, why the Introduction section presents the issue of food fortification). Afterwards the presented information must be associated with the direct aim and the model.
We added a few sentences in Lines 42-45 indicating the need for in vivo evaluation. We also indicated in this paragraph that supplementation, though effective, might lead to toxicity, especially when discussing vitamin D.
The model used serves to evaluate if the delivery vehicle, which is based on a stable nanoemulsions, is capable to enhance the status of vitamin D. Lines 106-107. We also added more information about the dose used in this model in Lines 242-249.
We have expanded our limitations paragraph (Lines 321-333) indicating that what the reviewer wants to know requires further evaluation. We are evaluating the ability of the carrier to deliver vitamin D in the context of vitamin D deficiency. We have not added this material into foods or supplements. We acknowledge this in the limitations: “PPI can be used to encapsulate other fat-soluble micronutrients and bioactives needed in fortification and supplementation programs such as vitamin A, E, and carotenoids. Nonetheless, the interaction of PPI alone and carrying nutrients with other ingredients and micronutrients within a more complex food matrix requires further evaluation. In addition, the potential influence of PPI carrying micronutrients on the sensory characteristics of foods is an important aspect that will dictate the success of this technology in food fortification programs.”
Most certainly, moving this technology into whole foods will require further research, which we are getting ready to conduct.
Materials and methods:
If the distribution is normal, Authors should state it.
We have indicated this in the text Lines 158-159. Additionally, this is the test of normality tables for a sample of our variables which is the main objective of this study
Tests of Normality | ||||||
Kolmogorov-Smirnova | Shapiro-Wilk | |||||
Statistic | df | Sig. | Statistic | df | Sig. | |
Nano_VitD_25OHVD_A | .243 | 7 | .200* | .890 | 7 | .274 |
Oil_VitD _25OHVD_A | .212 | 7 | .200* | .930 | 7 | .553 |
Nano_control_25OHVD_A | .244 | 7 | .200* | .946 | 7 | .694 |
Oil_control_25OHVD_A | .272 | 7 | .126 | .862 | 7 | .157 |
Control_25OHVD_A | .177 | 7 | .200* | .937 | 7 | .616 |
*. This is a lower bound of the true significance. | ||||||
a. Lilliefors Significance Correction | ||||||
Tests of Normality | ||||||
Kolmogorov-Smirnova | Shapiro-Wilk | |||||
Statistic | df | Sig. | Statistic | df | Sig. | |
Nano_VitD_25OHVD_A | .243 | 6 | .200* | .919 | 6 | .496 |
Oil_VitD _25OHVD_A | .225 | 6 | .200* | .900 | 6 | .374 |
Nano_control_25OHVD_A | .321 | 6 | .053 | .824 | 6 | .096 |
Oil_control_25OHVD_A | .179 | 6 | .200* | .922 | 6 | .518 |
Control_25OHVD_A | .200 | 6 | .200* | .908 | 6 | .423 |
*. This is a lower bound of the true significance. | ||||||
a. Lilliefors Significance Correction | ||||||
Results:
Table 1 – Authors should indicate not only dietary vitamin D intake but also total supply – from diet with supplementation added.
We have modified table 1 to include the information requested. The total supply per day cannot be calculated for the gavage dose as it was not given every day.
Apart from control group, all the remaining groups continue feeding on the same VitD deficient diet which add a neglectable amount of VitD if calculated per rat during the treatment period (about 1-2 IU/rat/week). The aim of the work is to compare the 2 equal doses of VitD by two types of carriers. The dietary VitD is neglectable and a common bias since all treatment groups received the same VitD-deficient diet. This clarification was added in lines 123-125.
Figure 3(B), Figure 4(B) – should be presented rather as tables to be easier to follow
We respectfully disagree with the reviewer. We argue that the figures are better than the tables to display this information. However, we have included this information in supplementary materials.
Discussion:
Authors should extensively address the short period of the study – Authors analysed the results after only one week of intervention – they should discuss why such a short intervention was planned and how it may have influenced the observed results
We have included a rationale for the short period of time in Lines 242-249. We followed a different study but decided to maintain a short time so as to reduce potential toxicity.
It seems that a number of factors were not analysed at baseline, but just after intervention – it should be extensively explained and discussed.
You are correct. We did not analyzed the same outcomes at baseline as for the end of the study. Our main outcome variable was circulating vitamin D. Furthermore, all rats were taken just after weaning a reared in the study lab exposed to the same factors except the diet and the intervention.
Author Contributions:
The section is shabbily prepared
We addressed this section in the previous review.
X.X. (line 304) – who is he?
This was a typo to hold a person in place, not initials. However, this was corrected in the previous version.
M.F.E. – it seems that he did not participated in the study at all
As stated in this section, this author worked on data collection and applying the methodology. It is included in the authorship.
Authors who did not participate in the manuscript preparing should be removed (due to the risk of guest authorship procedure that is forbidden) and presented in Acknowledgements Section
Understood.
Submission Date
12 November 2018
Date of this review
11 Dec 2018 22:36:48

Round 3
Reviewer 1 Report
The manuscript entitled “Stabilization of Vitamin D in Pea Protein Isolate Nanoemulsions Increases its bioefficacy in Rats” presents interesting issue. Moreover, Authors finally corrected majority of issues that were not corrected previously. And now only minor corrections are needed.
Abstract:
In the current version, the study is justified but there is no aim presented – Authors should present both of them: justification and the aim.
Results:
Figure 3(B), Figure 4(B) – should be presented rather as tables to be easier to follow
Discussion:
Authors should extensively address the short period of the study – Authors analysed the results after only one week of intervention – they should discuss why such a short intervention was planned and how it may have influenced the observed results.
It seems that a number of factors were not analysed at baseline, but just after intervention – it should be extensively explained and discussed.
Author Response
Dear Reviewer,
Thanks for taking the time to offer a fair review. We have bolded your comments in black for easy reading. Our responses are in blue in this document and in the revised version of the manuscript. We hope this review helps clarify your questions.
The manuscript entitled “Stabilization of Vitamin D in Pea Protein Isolate Nanoemulsions Increases its bioefficacy in Rats” presents interesting issue. Moreover, Authors finally corrected majority of issues that were not corrected previously. And now only minor corrections are needed.
Thanks for your efforts
Abstract:
In the current version, the study is justified but there is no aim presented – Authors should present both of them: justification and the aim.
Aim is added in line 15-17
Results:
Figure 3(B), Figure 4(B) – should be presented rather as tables to be easier to follow
Okay it is done
Discussion:
Authors should extensively address the short period of the study – Authors analysed the results after only one week of intervention – they should discuss why such a short intervention was planned and how it may have influenced the observed results.
Thanks for your comments. We made some changes in the manuscript to fix this limitation. Please refere to text line 107-108, 242-243, 254-257, 339-340
It seems that a number of factors were not analysed at baseline, but just after intervention – it should be extensively explained and discussed.
Based on our main objective which is bioefficacy of VitD, we measured 25OHVitD the baseline and after treatment period. This missed measurement of this baseline parameters is added as a limitation to this study. Please see lines 348-349.
Submission Date
12 November 2018
Date of this review
24 Dec 2018 22:36:48
